# CO_2_ Levels Modulate Carbon Utilization, Energy Levels and Inositol Polyphosphate Profile in *Chlorella*

**DOI:** 10.3390/plants12010129

**Published:** 2022-12-27

**Authors:** María Morales-Pineda, Maria Elena García-Gómez, Rodrigo Bedera-García, Mercedes García-González, Inmaculada Couso

**Affiliations:** Microalgae Systems Biology and Biotechnology Research Group, Institute for Plant Biochemistry and Photosynthesis, Universidad de Sevilla—Consejo Superior de Investigaciones Científicas, 41092 Seville, Spain

**Keywords:** green algae, carbon metabolism, photosynthetic capacity, lipids, ATP levels

## Abstract

Microalgae have a growing recognition of generating biomass and capturing carbon in the form of CO_2_. The genus *Chlorella* has especially attracted scientists’ attention due to its versatility in algal mass cultivation systems and its potential in mitigating CO_2_. However, some aspects of how these green microorganisms respond to increasing concentrations of CO_2_ remain unclear. In this work, we analyzed *Chlorella sorokiniana* and *Chlorella vulgaris* cells under low and high CO_2_ levels. We monitored different processes related to carbon flux from photosynthetic capacity to carbon sinks. Our data indicate that high concentration of CO_2_ favors growth and photosynthetic capacity of the two *Chlorella* strains. Different metabolites related to the tricarboxylic acid cycle and ATP levels also increased under high CO_2_ concentrations in *Chlorella sorokiniana*, reaching up to two-fold compared to low CO_2_ conditions. The signaling molecules, inositol polyphosphates, that regulate photosynthetic capacity in green microalgae were also affected by the CO_2_ levels, showing a deep profile modification of the inositol polyphosphates that over-accumulated by up to 50% in high CO_2_ versus low CO_2_ conditions. InsP_4_ and InsP_6_ increased 3- and 0.8-fold, respectively, in *Chlorella sorokiniana* after being subjected to 5% CO_2_ condition. These data indicate that the availability of CO_2_ could control carbon flux from photosynthesis to carbon storage and impact cell signaling integration and energy levels in these green cells. The presented results support the importance of further investigating the connections between carbon assimilation and cell signaling by polyphosphate inositols in microalgae to optimize their biotechnological applications.

## 1. Introduction

Increasing CO_2_ concentrations are accumulating in the lower layers of the atmosphere, causing the well-known greenhouse effect and subsequently global warming [1]. CO_2_ is the largest contributor, being responsible for up to 60 percent of the total greenhouse gases [2]. The major CO_2_ sinks on Earth are placed in oceans and large water bodies, where CO_2_ fixation via green microorganisms takes place. Microalgae perform photosynthesis efficiently, which transform CO_2_ to organic compounds without extra energy consumption [2,3]. Their metabolic plasticity allows them the ability to grow in big scale systems. The current focus is on microalgae as feedstock for bioenergy production, which are also a promising source to compensate and balance the increasing demands for biofuels, food, feed and valuable compound production [4,5,6].

Although carbon fixation is currently one of the most attractive features of algal biomass production, microalgae have additional benefits such as a high photosynthetic capacity, a rapid growth rate and excellent environmental adaptability that positively impact operational costs. Compared to plants, microalgae have several advantages for their use in sustainability plans, especially concerning ethical implications on food production or arable land use [7]. However, the efficiency of CO_2_ fixation and biomass production by microalgae largely depends on the cultivation conditions (media, temperature, light, pH, or nutrient availability), species differentiation, and the CO_2_ concentration, among others [2].

Among the microalgal species that have been used for different applications, *Chlorella* strains stand out for several reasons, such as the ability to fix carbon dioxide efficiently and to remove excess nitrogen and phosphorus in wastewater treatments [8,9]. Although *Chlorella* strains have been proposed as good candidates for biological mitigation of CO_2_ [10,11], specific information on how different CO_2_ levels impact carbon flux and algal metabolism is still missing.

In a previous study, five different genera from Chlorophyta (*Chlamydomonas, Chlorella, Scenedesmus, Monoraphydium* and *Chlorococcum*) were analyzed, and among them, Chlorella exhibited the highest values for growth rates and biomass accumulation, reaching 8.0–8.5 g L^−1^ in strains such as *Chlorella vulgaris* or *Chlorella sorokiniana* [12]. In this sense, carbon storage in the form of lipids and carbohydrates have been evaluated in different *Chlorella* strains, especially in studies dealing with nitrogen-starved cultures [13]. However, these results have been contradictory, as either carbohydrate [14,15] or lipid accumulation [16] was observed. In contrast, phosphorus storage has been only evaluated as polyphosphates [17] but has never been linked to phytic acid or other inositol polyphosphates (InsPs) in *Chlorella*.

InsPs are phosphorylated molecules that derive from the six-carbon-ring sugar myo-inositol, with great chemical complexity that has made it difficult to understand their biological role in eukaryotic cells. The identification and quantitation of the different chemical isomers have usually been linked to the use of radioisotopes until the application of LC-MS/MS methodology developed for the green model alga *Chlamydomonas* [18]. After using different structural and genetic analyses [19], they are now considered as signaling molecules that have an enormous impact on cell metabolism and energy levels in eukaryotes. InsP_6_ (also known as phytic acid) is a common way of P storage in plant seeds [20]. In algae, InsPs have been found to synergistically coordinate with the master cell growth regulator TOR (target of rapamycin) [18]. The TOR kinase is widely conserved in all eukaryotes and has been previously described in the model green alga *Chlamydomonas reinhardtii* [21]. InsPs are as widely conserved as TOR, and their role in green cells was firstly reported as important controllers of central carbon metabolism and carbon storage [18]. In a more recent study, they have been reported to have a major role in CO_2_ uptake, as they largely influence phosphorylation patterns of photosystem II (PSII) stabilization and assembly related proteins [22]. Oxygen evolving complex and electron transfer activity were also found aberrantly regulated in an InsPs biosynthetic mutant in the green model microalga *Chlamydomonas reinhadtii* [22]. All these data indicate that InsPs are essential components in the regulation of photosynthesis and carbon uptake that must also be monitored in other green microorganism in order to further evaluate carbon metabolism and storage.

In order to obtain a deeper understanding on how green cells adapt to high CO_2_ conditions, we evaluated *Chlorella sorokiniana* and *Chlorella vulgaris* growth as well as photosynthesis and carbon storage capacity under air and air supplemented with 5% CO_2_. Additionally, the level of different InsPs was analyzed to investigate the effect on these highly phosphorylated molecules that connect phosphorus storage and the regulation of carbon assimilation in algal cells. Finally, in order to picture how algal metabolism adapts to different carbon availabilities, we used *Chlorella sorokiniana* to analyze the levels of metabolites related to carbon metabolism, redox balance and energy levels under different CO_2_ concentrations.

## 2. Results

### 2.1. Chlorella Is Able to Grow under Different CO_2_ Conditions

In order to evaluate the effects of different CO_2_ concentrations on *Chlorella’s* growth, we chose two *Chlorella* strains that have been widely used in biotechnological applications and that have previously been used in studies connected with different environmental problems [23,24,25,26,27]. These *Chlorella* strains were then cultivated at 25 °C and 100 μE m^−2^ s^−1^ in Arnon medium under different CO_2_ concentrations (0, 1, 3 and 5%). The initial concentration for the starting cultures was OD_750nm_ 0.05 ± 0.01. The effects of different CO_2_ concentrations on the growth of the two strains are shown in Supplemental Appendix A, panels A to D. *Chlorella sorokiniana* reached the highest cell density under 1% CO_2_ condition, and other CO_2_ concentrations (3 and 5%) did not affect its growth. However, *C. vulgaris* did not reach the highest cell density until 3% of CO_2_ and showed lower cell density than *C. sorokiniana* under low CO_2_. In contrast, both strains behave similarly under 3 and 5% CO_2_, both reaching a 0.06 specific growth rate (μ) (Table 1).

### 2.2. Photosynthetic Activity Is Different between Chlorella Strains

The evaluation of green algae as a good sink of CO_2_ involves studying the photosynthetic capacity of these microorganisms. In this study, the maximum quantum yield of PSII (Fv/Fm) was measured by the saturating pulse method using a pulse–amplitude modulation (PAM) fluorimeter [28,29] in dark-adapted cultures at a mid-log phase growth. The values of the Fv/Fm ratio in the different cultures bubbling with air were around 0.8 in *C. sorokiniana*, and these values were kept virtually constant along the increasing concentrations of CO_2_ supplied (Table 1). However, Fv/Fm in *C. vulgaris* increased from 0.79 to 0.86 (Table 1). In order to compare how electron transfer rate (ETR) responds under air and 5% CO_2_, we subjected both strains to a light curve reaching 2000 μE m^−2^ s^−1^ using PAM (Figure 1A,B). The data showed an increase of 10% in *C. sorokiniana* (Figure 1A) and 20% in *C. vulgaris* in the presence of high CO_2_ compared to air bubbling (Figure 1; panel B). *C. vulgaris* reached the maximum ETR under lower irradiance (344 μE m^−2^ s^−1^) (Figure 1B) versus *C. sorokiniana* that reached 50 R.U. of ETR after illumination with 536 μE m^−2^ s^−1^ (Figure 1A) under 5% CO_2_. In contrast, when subjected to air condition, *C. sorokiniana* reached the maximum ETR at 344 μE m^−2^ s^−1^ with 48 R.U., while *C. vulgaris* showed a reduction in the maximum ETR (39 R.U.) upon the same illumination and air conditions.

In order to evaluate the capacity of these strains to dissipate the excess of energy under the different concentrations of CO_2_, we evaluated their capacity for regulated dissipation (Y(NPQ)) and non–regulated dissipation of energy (Y(NO)) (Figure 1D and Appendix A). We found that *C. sorokiniana* and *C. vulgaris* both showed very low levels for Y(NPQ) with maximums of 0.21 and 0.16, respectively, at the highest light intensity (2000 μE m^−2^ s^−1^). In the case of C. vulgaris, a big decrease in Y(NPQ) was seen at 5% supplemented air condition. The opposite was seen in Y(NO) values, where C. sorokiniana and *C. vulgaris* showed similar levels under air (Figure 1E,F), but *C. vulgaris* increased Y(NO) at 5% CO_2_, most likely to compensate for low levels of Y(NPQ) in these conditions (Figure 1D).

### 2.3. Chlorella Strains Accumulate InsPs and Orthophosphate under 5% CO_2_ Concentration

InsPs in green algae were only examined in the genus *Chlamydomonas* as intertalkers with the TOR signaling pathway [18]; however, the role of these molecules in other microalgae has not yet been examined. In this study, we measured the profile of the different InsPs present in *Chlorella* strains under two conditions: air and 5% CO_2_. All the samples were normalized by weight, and an internal standard (1 μM 3–fluoro–InsP_3_) was added for controlling for possible sample loss. We used the LC–MS/MS technique described in the Methods section and previously reported in Couso et al. (2016) [18] to measure InsPs. We found detectable levels for InsP_3_, InsP_4_, InsP_5_ and InsP_6_ (also known as phytic acid); however, we could not detect pyro–phosphorylated forms in contrast to *Chlamydomonas* InsPs profile. Both *Chlorella* strains tended to accumulate InsP_6_ (phytic acid) under low levels of CO_2_ (Figure 2A,B), but in the presence of 5% supplemented CO_2_, their profile changed significantly. While *C. sorokiniana* tended to accumulate InsP_4_ and InsP_6_, *C. vulgaris* showed significant differences only in InsP_6_ levels. *C. sorokiniana* showed very low levels of InsP_5_ compared to *C. vulgaris*, indicating different regulation in the biosynthesis of these compounds (Figure 2A,B). In addition, total levels of InsPs were quantified (Figure 2C), and the comparison between CO_2_ conditions showed an important increase under 5% CO_2_ versus air in both *Chlorella* strains. This increase was especially significant in the case of *C. sorokiniana* that reached 2.13 times the level under control conditions, compared to 1.48 times in the case of *C. vulgaris* (Figure 2C).

Orthophosphate levels were also quantified under the same conditions (air and 5% CO_2_) in both strains (Figure 2D). Although *C. sorokiniana* showed smaller levels of PO_4_^−3^ than *C. vulgaris* under control conditions, they both had increased PO_4_^−3^ levels under 5% CO_2_ (Figure 2D), showing very similar levels in both strains.

### 2.4. Chlorella sorokinana and Chlorella vulgaris Differentially Distribute Carbon Storage under 5% CO_2_ Condition

Lipid droplets are the major site of neutral lipid storage reported in algal cells, and they are positively correlated with their lipid content, which is important for their consideration as biofuel feedstocks [30,31]. In our study, we compared the lipid bodies accumulated in the two *Chlorella* strains under air and 5% CO_2_ supplemented air. *Chlorella* cells were dyed using Nile red and were visualized under a fluorescent microscope as described in the Methods section (Figure 3A). We observed that both *C. sorokiniana* and *C. vulgaris* showed a very similar number of lipid bodies under no CO_2_ added and for both accumulated lipid bodies under the supplemented CO_2_ condition, as shown in Figure 3B. ImageJ quantitation revealed that *C. sorokiniana* accumulated around 20% more lipid bodies than *C. vulgaris* in these conditions (Figure 3B). The results indicate that 5% CO_2_ supplementation was a worth–trying condition that must be beneficial for the use of these microalgae for biotechnological purposes in the field of biofuel production.

In order to completely evaluate potential carbon sinks in these strains, starch quantitation was also performed in the same conditions in both strains. The Total Starch Assay kit (Megazyme) was used to measure the starch levels in the two *Chlorella* strains under air bubbling and 5% CO_2_ supplemented air (Figure 3C). Under air conditions, starch levels were very similar in both strains; however, under 5% CO_2_ conditions, they behaved the opposite. *C. vulgaris* tended to increase starch levels, while *C. sorokiniana* tended to decrease these levels. These data indicate that carbon flux and especially carbon sinks are differentially regulated between these strains.

### 2.5. Metabolic Adaptation in Chlorella sorokinana under Different CO_2_ Conditions

The information about how CO_2_ influences algal metabolism is still very limited. In this study, different metabolites were measured under different concentrations of CO_2_ (air, air supplemented with 5% CO_2_) in *C. sorokiniana*, as it was better adapted to different levels of CO_2_ conditions than *C. vulgaris*. Different metabolites related to Calvin cycle/glycolysis, the tricarboxylic acid cycle (TCA) and energy levels were quantified by using ultra–performance liquid chromatography (UPLC/MS) determination (Figure 4; all panels).

Overall, the Calvin cycle/glycolysis–related metabolites (Figure 4A) increased under 5% CO_2_, except from fructose-6-phosphate (F6P), which showed a non–significant increase in this condition. Instead, the level of glucose 6-phosphate (G6P) increased 40%, 3–phosphoglicerate (3PG) increased 56%, and phosphoenolpyruvate (PEP) increased 50% after being subjected to 5% CO_2_. The highest levels under 5% CO_2_ were observed in fructose-1,6-biphosphate (F1,6P) that reached over six-fold the level under air condition, or dihydroxyacetone-phosphate (DHAP) that reached 1.6-fold the level under the same conditions. Overall, these increases indicate the acceleration of glycolysis after supplementation with CO_2_. In this sense, we further analyzed the levels of metabolites related to the TCA cycle from early steps (acetyl CoA, CoA) to organic acids (succinic, fumaric, malic and oxaloacetic) (Figure 4B). Acetyl CoA and CoA both highly increased after CO_2_ supplementation, reaching 13- and 5-fold the levels in air, respectively (Figure 4B, small panels). Fumaric and malic acids also followed the same trend, almost doubling their levels in supplemented conditions (Figure 4B). In contrast, succinic acid showed an important decrease (34%) under supplementation with CO_2_. These data indicate an important redistribution of the organic acids dependent on CO_2_ concentration and reflect the versatility of this microalgae–incorporating carbon.

In order to evaluate how energy levels of the algal cells respond to increasing concentrations of CO_2_, we measured AMP, ADP and ATP levels using the same conditions. *C. sorokinina* showed important increases in the three nucleotide phosphorylated forms when subjected to 5% CO_2_. ATP and ADP levels doubled under air conditions while AMP increased by 48% under 5% CO_2_. These data together with the increased levels of glycolysis intermediates and TCA metabolites indicate an activation of metabolism during supplementation of CO_2_ that is also reflected by higher growth (Appendix A).

The redox balance is a good indicator of the stressful conditions tested in the algal cells. Glutathione (GSH; γ–glutamyl–cysteinyl–glycine) is considered as a non-enzymatic antioxidant that is widely distributed in most plant tissues. GSH takes part in the detoxification of ROS, directly or indirectly [32,33], and it is converted into glutathione disulfide (GSSG) by the enzyme GPX. GSSG can be reconverted/recycled again into GSH by the activity of GR [34] coupling with NADP+. In these experiments, GSH and GSSG levels increased in the presence of 5% CO_2_ compared with air conditions (Figure 5A,B). The ratio GSH:GSSG showed an important increase (two-fold) under CO_2_ supplemented compared to air conditions. The same happened with NADP and NADPH that showed an important increase of 2.1- and 2.5-fold, respectively (Figure 5C,D).

## 3. Discussion

The genus *Chlorella* has received increasing attention due to its easy and rapid growth for broad industrial applications [35]. Due to a complicated taxonomy, the term “*Chlorella*” has referred to a spherical cell phenotype including the class Chlorophyceae and Trebouxiophyceae [36,37,38]. After genome sequencing and “omics” data availability, it is possible to make a fair comparison of two organisms that fall into the same phylogenetic group such as *Chlorella sorokiniana* and *Chlorella vulgaris* [37]. Both Chlorella strains have previously been used in different studies, evaluating their growth under high CO_2_, reaching 50% [39,40,41] to optimize lipid production and to boost CO_2_ biofixation using these algae.

In our study, we used moderate levels of CO_2_ in order to evaluate the gradual adaptability of these cells to increasing concentrations of CO_2_ and to avoid any effects on the pH of the media culture. After subjecting *Chlorella* cultures to different concentrations of CO_2_ (air to 5%), differences in growth rates were observed, especially under air where *C. sorokiniana* better adapted to CO_2_ conditions than *C. vulgaris* (Appendix A). Fv/Fm were constant under these conditions in *C. sorokiniana* while *C. vulgaris* increased its photosynthetic capacity when subjected to increasing CO_2_ concentrations (Table 1). This was also observed in other *Chlorella* strains such as *C. pyrenoidosa* that adapted Fv/Fm to increasing CO_2_ in contrast to *Chlamydomonas reinhardtii* that tended to decrease this parameter under high CO_2_ [42].

Significant differences were found in the levels of ETR when comparing the two *Chlorella* strains under a light curve. In *C. sorokiniana*, the maximum ETR did not change between the two CO_2_ conditions tested (air and 5% CO_2_); however, this alga reached maximum ETR at lower illumination under air conditions versus 5% CO_2_. *C. vulgaris* has a lower maximum ETR under air conditions than *C. sorokinina*, but it reached maximum ETR at the same light intensity under the two CO_2_ conditions (Figure 1A,B). These results suggest that illumination and CO_2_ levels control photosynthesis in different ways in these *Chlorella* strains. A recent study also reported that photosynthetic properties of *C. vulgaris* and *C. sorokiniana* are differently influenced by CO_2_ availability [41]. This information is very valuable for scaling up experiments in order to maintain efficient photosynthetic activities under various CO_2_ conditions. The response of photoautotrophic algal cultures to variations in CO_2_ conditions was also studied in *Chlorella variabilis* [43]. These results suggest the importance of photoprotective mechanisms, including NPQ, to maintain photosynthesis under various CO_2_ conditions. In this study, excess energy dissipation was performed in different ways to compare the two strains. While Y(NPQ) and Y(NO) were largely unaffected in *C. sorokinina*, *C. vulgaris* showed an increase in Y(NO) and subsequent decrease in Y(NPQ) under 5% CO_2_ (Figure 1B–E). These data further indicate a connection between CO_2_ levels and the photosynthetic activity in these green cells.

Inositol polyphosphates have recently emerged as highly phosphorylated molecules that have an important signaling role in green microalgae, especially related to carbon metabolism [18]. The InsPs profile also responds to carbon sources and the deficiency of high–order InsPs deregulates photosynthesis in *Chlamydomonas* [18,22]. However, how the InsPs profile responds to availability of CO_2_ was not previously reported in any green organism. Here, we found that both *Chlorella* tend to accumulate InsPs, especially in the form of phytic acid (InsP_6_), nearly reaching two-fold in the case of *C. sorokiniana* under 5% CO_2_ (Figure 2B). These results connect InsPs biosynthesis and CO_2_ assimilation in *Chlorella*, as it was also reported in *Chlamydomonas* [18]. Apart from this, the InsPs profiles are somehow different between the two strains, as *C. sorokinina* increased three-fold the level in InsP_4_ under 5% CO_2_ while *C. vulgaris* did not. In addition, total levels of InsPs were 20% higher in *C. sorokinina* than in *C. vulgaris*, indicating that the InsPs biosynthetic pathway has higher activation in the first strain (Figure 2C). In contrast, total phosphate levels were not significantly different between both strains under 5% CO_2_ (Figure 2D), although we found significant increases in both strains when comparing air and 5% CO_2_ conditions (Figure 2D). These data suggest that CO_2_ controls InsPs biosynthesis, and these molecules can also contribute to phosphate storage increases in these green cells.

InsPs have also been linked to the regulation of carbon storage in the form of lipids in microalgae [18]. In this study, lipid bodies were monitored in *C. sorokiniana* and *C. vulgaris* using Nile red staining (Figure 3A). We found an important increase in the accumulation of lipid bodies under 5% CO_2_ conditions in both strains that was especially significant in *C. sorokiniana* (Figure 3B). On the contrary, in this strain, we found a significant decrease in starch after subjecting it to 5% CO_2_ conditions (Figure 3C). Lipids and starch are normally considered as competing carbon sinks in green microalgae [44,45], and this could partially explain this decrease, but also the differences found in InsPs levels could affect lipid accumulation in *C. sorokiniana*, as previously seen in *Chlamydomonas* [18]. This increased contrast with *C. vulgaris*’ starch levels that did not significantly change following 5% CO_2_ conditions (Figure 3C) further suggests a different regulation of carbon utilization by both microalgal strains.

After analyzing our data on the two *Chlorella* strains, we decided to further investigate the effect of CO_2_ on carbon metabolism in *C. sorokiniana*, as it seemed to better adapt to the CO_2_ concentrations tested. We analyzed different metabolites related to Calvin and TCA cycles, and we evaluated phosphorylated nucleotides including ATP. *C. sorokiniana* showed an important increase in Calvin–related metabolites such as F6P, 3-PG or PEP that are also related to glycolysis (Figure 4A) and TCA–related metabolites such as malic or fumaric acid under 5% CO_2_ conditions. These suggest a boost in the 5% CO_2_ carbon assimilation in this alga and an increase in the energetic charge of these cells. After a transcriptome analysis, a similar response was seen in the highly CO_2_–tolerant *Chlorella sp*. in contrast to the low–tolerant strain showing downregulation of these pathways [46].

The glutathione cycle is a well–known antioxidant process that green organisms use to detoxify ROS and avoid oxidative stress. After analyzing GSH and GSSG levels under air and 5% CO_2_ conditions, we found the GSH/GSSG ratio to dramatically increase together with NADP and NADPH. Our data suggest that this microalga needs redox rebalancing under CO_2_ conditions in order to keep cell homeostasis after the increases in carbon assimilation and photosynthetic performance. This redox balance has been observed in *Chlorella* cells under high light stresses as part of the photoacclimation process [47], but our data indicate that CO_2_ levels can also activate this process in green cells.

## 4. Conclusions

Our data suggest that *Chlorella* adapts its photosynthesis capacity and photoprotection and enhances its metabolism to increase the production of lipids to cope with increasing CO_2_ concentrations. We also show that the InsPs profile adapts to CO_2_ availability, something that has not been reported before in green organisms. These data further indicate a connection between InsPs regulation and carbon flux in these green organisms that is important to understand for future biotechnological applications of these green microalgae either in carbon biomitigation and/or biofuel production.

## 5. Materials and Methods

### 5.1. Strains and Growth Conditions

*Chlorella sorokiniana* UTEX 1230, *Chlorella vulgaris* UTEX 2714 were obtained from the Algae Culture Collection at the University of Texas. These green microalgae were grown photoautotrophically in Allen and Arnon medium [48], at 25 °C. The liquid cultures were continuously bubbled with air (approx. 0.04% CO_2_) and air supplemented with 1, 3 or 5% (*v*/*v*) CO_2_ as the only source of carbon (Appendix A). Cells were grown in Roux flasks of 1 L capacity, laterally and continuously illuminated with mercury halide lamps at 50 μE m^−2^ s^−1^. The light intensity was measured at the surface of the flasks using a LI–COR quantum sensor (model L1–1905B, Li–Cor, Inc., Lincoln, NE, USA). We measured the growth kinetics in different concentrations of CO_2_ (air, 1, 3 and 5%) of the two strains using mean values of OD 750 nm measurements performed in triplicate. Growth rate was calculated from: ln (N2 − N1)/t2 − t1 [49].

### 5.2. Pulse–Amplitude Modulation Fluorometry

Fluorescence of chlorophyll a was measured at room temperature using a pulse–amplitude modulation fluorometer (DUAL–PAM–100, Walz, Effeltrich, Germany). The maximum quantum yield of PSII was assayed after incubation of the algal suspensions in the dark for 15 min by calculating the ratio of the variable fluorescence, Fv, to maximal fluorescence, Fm (Fv/Fm). The parameters Y(NPQ) and Y(NO) corresponding to the quantum yield of PSII photochemistry were calculated by the DUAL–PAM–100 software according to the equations in [50,51]. Measurements of relative linear electron transport rates were based on chlorophyll fluorescence of pre-illuminated samples, applying stepwise increasing actinic light intensities up to 2000 μE m^−2^ s^−1^.

### 5.3. Metabolite Sample Preparation and Analysis

For metabolite content determination, *Chlorella sorokiniana* cell pellets were lyophilized (Skadi–Europe TFD 8503), flushed with a nitrogen stream to prevent oxidation, and stored at −20 °C. Primary metabolites were determined from 20 mg of lyophilized biomass subjected to mechanical disruption in a Mini Bead Beater (Biospe Products) with a mixture of 2.7 and 0.5 mm glass beads (ratio 1/3) in the presence of 1 mL extraction buffer consisting of chloroform:methanol (3:7, *v*/*v*). As internal standard, 40 μL of paracetamol 100 μM was added. Following centrifugation at 5000× *g* for 5 min at RT (room temperature), the supernatant was collected. This process was repeated, adding 1 mL of extraction buffer until the supernatant was colorless. The combined supernatants were dried under nitrogen stream, resuspended in Milli–Q water and submitted for analysis. Primary metabolite determination was carried out by ultra–high–performance liquid chromatography system coupled with mass spectrometry (UPLC/MS) as described in [52].

### 5.4. Inositol Polyphosphates Analysis and Orthophosphate Quantitation

The two *Chlorella* strains were grown either in air or in air supplemented with 5% CO_2_ (*v*/*v*). Samples for inositol determination were collected in exponential growth phase (1–2 × 10^6^ cells mL^−1^). The number of cells of the samples was adjusted so that all replicates had identical volume. After collecting *Chlorella* cells (4000 rcf, 5 min, room temperature), InsPs were extracted using 1 mL final volume 5% trichloroacetic acid and flash frozen in liquid nitrogen. The samples were centrifuged at a maximum speed in a microfuge at 4 °C for 20 min, and then, the supernatants were supplemented with 1 μM 3–fluoro–InsP_3_ (Enzo Life Sciences), which served as an internal standard for normalization. Samples were extracted three times with 2 mL of water–saturated diethyl ether to remove contaminants. The pooled aqueous phase from the extractions was loaded onto a Strata–X AW column (Phenomenex; 30 mg resin; weak anion mixed mode phase 33 mm particle size). The column was washed with 1 mL of 25% methanol to remove trichloroacetic acid and other contaminants, and the InsPs were eluted using 1 mL of 100 mM ammonium carbonate. Then, 0.5 mL of acetic acid was added to each eluate, and the samples were vacuum–dried. Each sample was resuspended in 50 μL of ultrapure water just prior to LC–MS/MS analysis. The final LC–MS/MS injection volume was 8 μL.

LC–MS/MS data acquisition was performed as described in [18]. Data were analyzed using the QualBrowser and QuanBrowser applications of Xcalibur (Thermo Fisher Scientific). Data were normalized using the internal standard 3–fluoro–InsP3.

Samples for orthophosphate determination were collected in exponential growth phase (1–2 × 10^6^ cells mL^−1^), and 50 mL cell pellets were used for the colorimetric determination using Phosphate Assay Kit (SIGMA) following manufacturer´s instructions.

### 5.5. Nile Red Staining and Fluorescence Microscopy

The two *Chlorella* strains were grown in air bubbling and 5% CO_2_ (as described above). Cells were fixed on ice for 20 min with 2% paraformaldehyde (Sigma–Aldrich, 158127) and then washed with PBS buffer twice. Lipid body staining was performed as described [53], adding an incubation step of the dye for 20 min at 37 °C. Microscopy was performed with a microscope DM6000B (Leica) using a ×100 oil immersion objective with DIC optics or wide–field fluorescence equipped with a Leica L5 filter cube (excitation bandpass 480/40 nm; dichroic 505 nm; emission bandpass 527/30 nm) and an ORCAER camera (Hamamatsu).

After visualization of lipid bodies using Nile Red staining, we used Image J (https://imagej.nih.gov/ij accessed on 21 June 2018) Particle Count Analysis on approximately 100 cells per strain and condition.

### 5.6. Starch Quantification

Starch was measured using a Total Starch Assay Kit (AA/AMG; Megazyme) following the manufacturer’s instructions but scaled down to 10 mg freeze–dried cell powder as starting material.

### 5.7. Statistical Analysis

Biological experiments from Figure 1, Figure 2 and Figure 3 were performed in triplicate with three technical replicates each, except for Image J analysis, which was performed on approximately 100 cells for each condition. Metabolite analysis in Figure 4 and Figure 5 were performed in quintuplicate for each condition, and two biological replicates were analyzed. Means and standard deviations (SDs) were then calculated for each sample analysis, and SDs are represented by error bars in all figures. Significant differences at *p* value < 0.05 were calculated according to Student’s *t* test.

## Figures and Tables

**Figure 1 plants-12-00129-f001:**
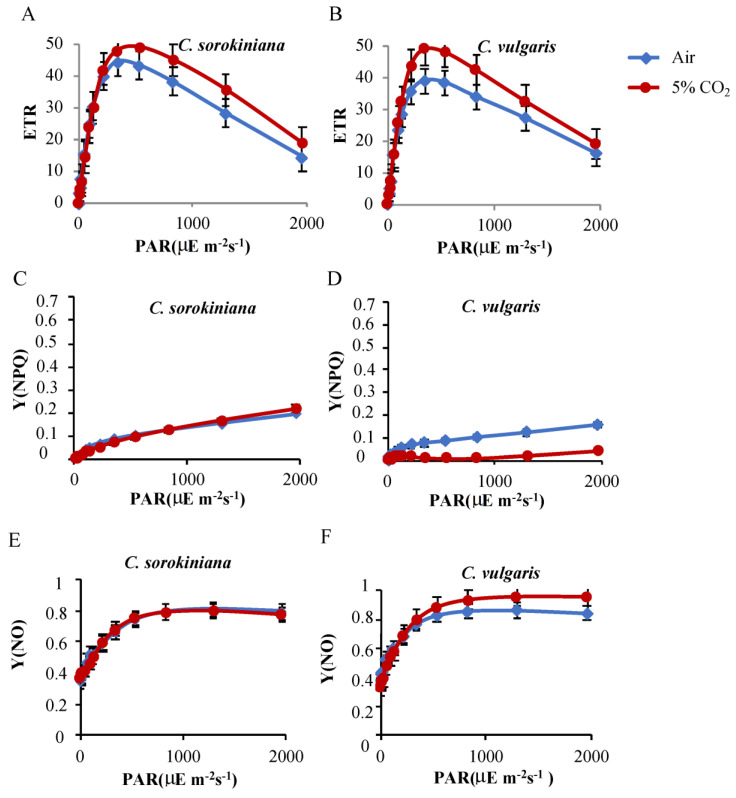
Photosynthetic activity of *Cholella* strains. (**A**,**B**) ETR curve evaluated under increasing actinic light (0–2000 μE m^−2^ s^−1^) of *Chlorella sorokinina* and *Chlorella vulgaris* tested under increasing CO_2_ (air–5% CO_2_). Data are the mean ± SE of three biological replicates performed in duplicate. (**C**,**D**) Light responses of Y(NPQ) in the two *Chlorella* strains under increasing CO_2_ (air 5%). (**E**,**F**) Y(NPQ) corresponds to the fraction of energy dissipated in the form of heat via the regulated non–photochemical quenching mechanism. The mean ± SE was calculated from three independent biological replicates performed in triplicate.

**Figure 2 plants-12-00129-f002:**
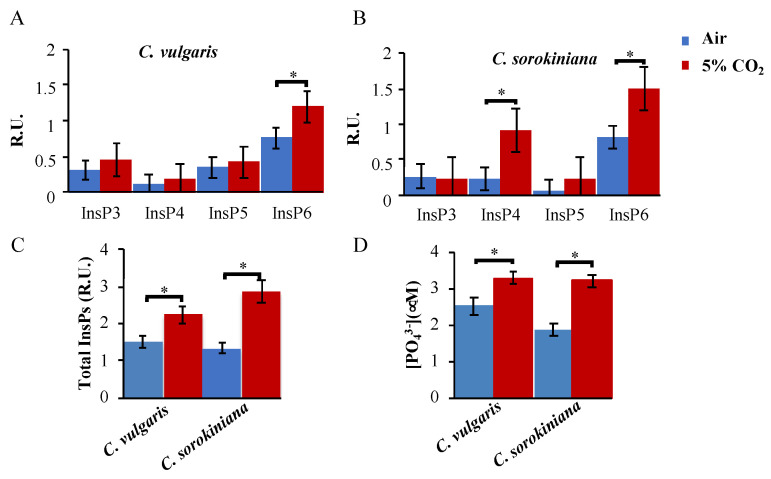
Accumulation of InsPs and orthophosphate under 5% CO_2_ concentration in *Chlorella* strains: (**A**,**B**) Inositol polyphosphate (InsP_3_, InsP_4_, InsP_5_ and InsP_6_) levels in the two Chlorella strains under air or 5% of CO_2_. (**C**) Total InsPs levels under the same conditions and strains. (**D**) Total orthophosphate concentration in cell samples of *C. sorokiniana* and *C. vulgaris* under the same mentioned conditions. The mean ± SE was calculated from three independent biological replicates performed in triplicate. The measurements were performed as indicated in the Methods section. * represent significant differences (*p* < 0.05) evaluated using Student’s *t* test.

**Figure 3 plants-12-00129-f003:**
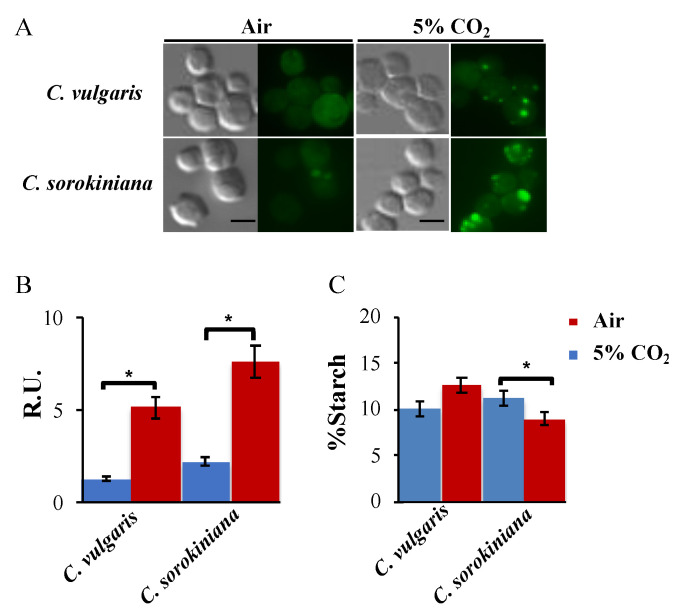
Carbon storage distribution in *Chlorella* strains: (**A**) Lipid bodies were stained with Nile red and imaged by differential interference contrast (DIC) and fluorescence microscopy in *the C. sorokiniana* and *C. vulgaris* either in air or 5% CO_2_. Scale bar = 8 mm. (**B**) Quantification of Nile red fluorescence (see Methods). R.U., relative units. (**C**) Starch levels of the reported strains under air and 5% CO_2._ The mean ± SE was calculated from three independent biological replicates performed in triplicate. The measurements were performed as indicated in the Methods section. * represent significant differences (*p* < 0.05) evaluated using Student’s *t* test.

**Figure 4 plants-12-00129-f004:**
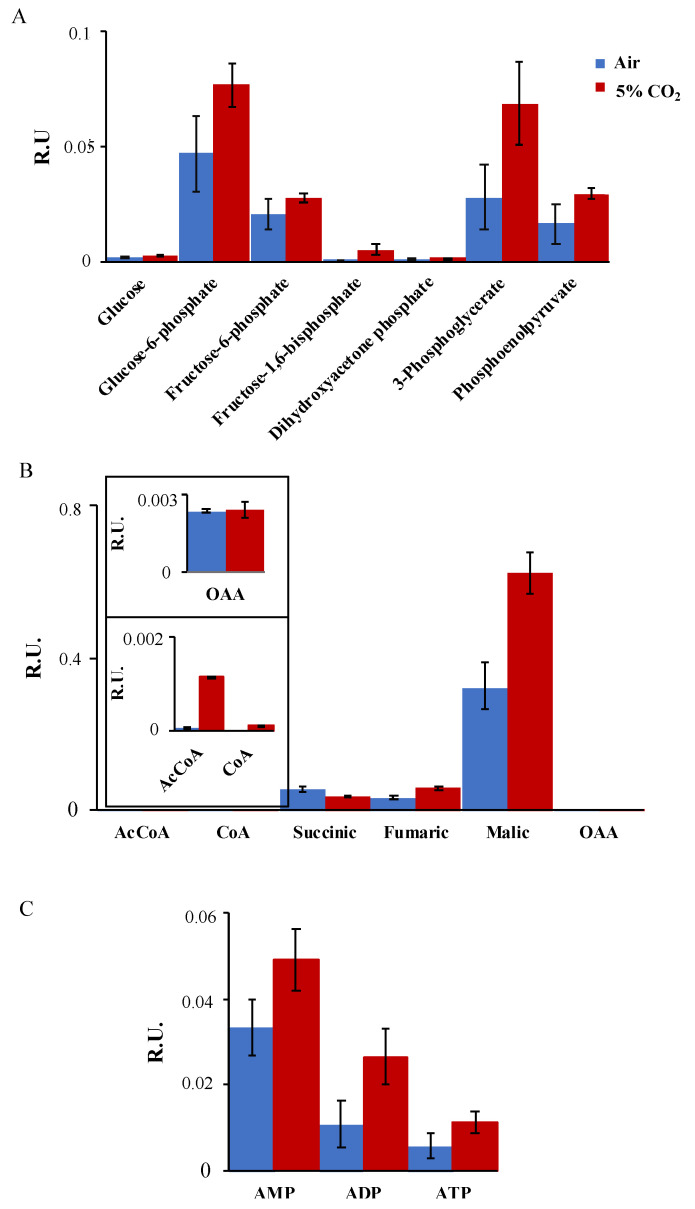
Metabolites presented in C. sorokiniana under different CO_2_ concentrations: UPLC–MS analysis of (**A**) glycolysis–related metabolites, (**B**) TCA related metabolites and (**C**) phosphorylated nucleotides in *C. sorokiniana* samples subjected to air or 5% CO_2_. The mean ± SE was calculated and performed in quintuplicate for each condition, and two biological replicates were analyzed as described in the Methods section.

**Figure 5 plants-12-00129-f005:**
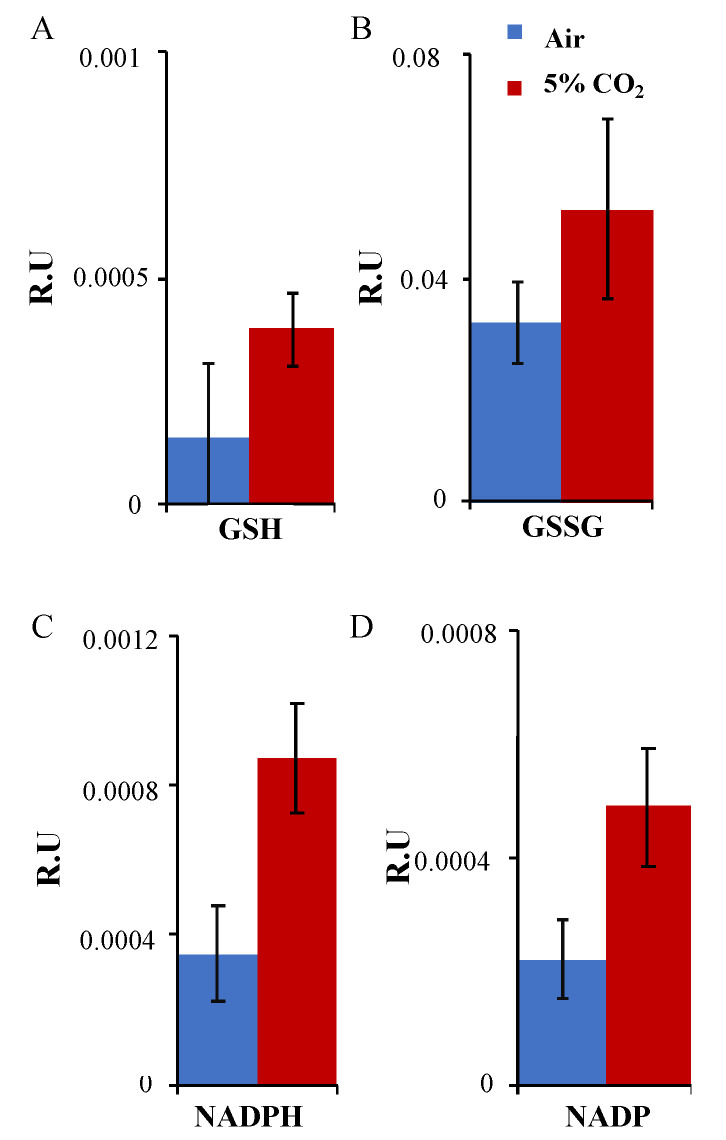
Redox balance: (**A**) Metabolite analysis of GSH; (**B**) GSSG; (**C**) NADPH; (**D**) NADP in *Chlorella sorokiniana* samples under air and 5% CO_2_. The mean ± SE was calculated and performed in quintuplicate for each condition, and two biological replicates were analyzed as described in the Methods section.

**Table 1 plants-12-00129-t001:** Specific growth rate μ (h^−1^) and the maximum quantum yield of PSII (Fv/Fm) calculated from *Chlorella* cultures grown in increasing CO_2_ concentrations (0% to 5% CO_2_).

	*C. sorokiniana*		*C. vulgaris*	
[CO_2_]	μ (h^−1^)	Fv/Fm	μ (h^−1^)	Fv/Fm
Air	0.06 ± 0.003	0.81 ± 0.03	0.05 ± 0.001	0.79 ± 0.03
1%	0.06 ± 0.002	0.81 ± 0.05	0.05 ± 0.002	0.85 ± 0.02
3%	0.06 ± 0.002	0.83 ± 0.04	0.06 ± 0.003	0.85 ± 0.02
5%	0.06 ± 0.005	0.81 ± 0.02	0.06 ± 0.002	0.86 ± 0.05

## Data Availability

Not applicable.

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
