# Peer review of "CO2 Levels Modulate Carbon Utilization, Energy Levels and Inositol Polyphosphate Profile in Chlorella"

_plants, 2022, doi:10.3390/plants12010129_

Round 1
Reviewer 1 Report
The topic is interesting but the MS needs major revisions before acceptance for possible publication in the ‘Plants’. The following issues need to be addressed to improve transparency and communication of the results:
Some comments and suggestions about MS
Title
Line 2-3: The title of manuscript is good.
Abstract
Line 16: Please replace 'concentrations' with 'concentration'.
Line 26: Please avoid repeating the same words (like Chlorella, Inositol Polyphosphates, Carbon, CO2) that you have already used in the title to enhance the visibility of your article.
Introduction
The introduction is fine.
Results
Line 104-105: A table should be self-explanatory and its title should be detailed. So, the Title of Table 1 should be improved. What is Fv/Fm? Please describe in the table’s title.
Line 168-172: Justify the caption of Figure 2.
Line 199-205: Same comments as for Lines 168-172.
Line 239-242: Same comments as for Lines 168-172.
Discussion
Line 259-338: The discussion is shallow and should be improved. Further, assign the numbers to subheadings or remove the subheadings in the Discussion section.
Materials and Methods
Please add the schematic diagram of the study for easy understanding..
Please write a detailed Statistical analysis under a separate heading.
Line 339-422: Assign the numbers to subheadings or remove the subheadings in the Discussion section.
Conclusions
Conclusions need to write on the basis of the key findings of your study for the easy understanding of readers.
Please take into consideration the comments on the PDF file of the revision.

Author Response
Response to Reviewer 1 Comments
Point 1: The topic is interesting but the MS needs major revisions before acceptance for possible publication in the ‘Plants’. The following issues need to be addressed to improve transparency and communication of the results:
Response 1: We want to thank reviewer 1 for giving us the opportunity to improve our manuscript and we really appreciate his/her comments.
Point 2: Abstract
Line 16: Please replace 'concentrations' with 'concentration'.
Line 26: Please avoid repeating the same words (like Chlorella, Inositol Polyphosphates, Carbon, CO2) that you have already used in the title to enhance the visibility of your article.
Response 2: We have replaced “concentration” and changed keywords to increase the visibility of our manuscript as suggested by reviewer 1.
Point 3: Results
Line 104-105: A table should be self-explanatory and its title should be detailed. So, the Title of
Table 1 should be improved. What is Fv/Fm? Please describe in the table’s title.
Line 168-172: Justify the caption of Figure 2.
Line 199-205: Same comments as for Lines 168-172.
Line 239-242: Same comments as for Lines 168-172.
Response 3: We have changed the title in Table I including the definition of Fv/Fm. We used this measurement to evaluate the photosynthetic capacity of the two Chlorella strains used in this study. We have also justified the text in figure legends as suggested by reviewer 1.
Point 4: Discussion
Line 259-338: The discussion is shallow and should be improved. Further, assign the numbers to subheadings or remove the subheadings in the Discussion section.
Response 4: We have revised the discussion, analyzed the different results we obtained deeper, and updated the references following reviewer 1 comments. We have also removed the subheadings as suggested by reviewer 1.
Point 5: Materials and Methods
Please add the schematic diagram of the study for easy understanding..
Please write a detailed Statistical analysis under a separate heading.
Line 339-422: Assign the numbers to subheadings or remove the subheadings in the Discussion section.
Response 5: We have added a supplemental figure indicating the global procedure of the manuscript as suggested by reviewer 1. We have added a statistical analysis section to the “Material and methods” section and included the subheadings.
Point 6: Conclusions
Conclusions need to write on the basis of the key findings of your study for the easy understanding of readers
Response 6: We have added a final conclusion section as suggested by reviewer 1.
Point 7: Please take into consideration the comments on the PDF file of the revision. 2
Response 7: We appreciate that reviewer 1 suggests new references and we have added one of the suggested references to the introduction section.

Reviewer 2 Report
The manuscript "CO2 levels modulate carbon utilization, energy levels, and inositol polyphosphates profile in Chlorella" deals with the understanding of how green cells adapt to high CO2 conditions. The paper is well organized and well written, covering the most significant topics in the field. However, I have the following comments to be considered through a MAJOR REVISION;
Point 1: In the abstract, there is no mention of any quantitative findings at all. It is recommended to the writers that they add some quantitative findings in order to improve the readability and comprehension of the paper.
Point 2: Why does the author only estimate the impact of CO2 to be 5%?
Point 3: How can we determine whether or not a CO2 concentration of 5% meets the criterion for being deemed high?
Point 4: The innovation of this study must be emphasized in the introduction.
Point 5: Line 90-91, This phrase is unacceptable to me due to the fact that the CO2 tolerance ability of microalgae was changed not only by the concentration of CO2 but also by the flow rate of CO2. In addition, a number of studies have shown that Chlorella can thrive in environments with high concentrations of CO2 ranging from 10 to 40%.
Point 6: The authors should include a concise commentary on the drawbacks of the systems that were investigated. To provide the reader with a deeper understanding of the topic, it is recommended that one whole paragraph be included.
Point 7: When writing the abstract, the body of the text, the table, or the figure, you should avoid using abbreviations before explaining what they mean.
Point 8: References need update: Although the number of references is already very high, further recent relevant references from the past 5 years should be cited. Please check if you really need all those references cited.
Point 9: The manuscript cites 59 references, none of which were published in the PLANTS, suggesting that it is not of direct relevance to PLANTS readers. Consider also citing reference(s) published in PLANTS.
Author Response
Response to Reviewer 2 Comments
Point 1: In the abstract, there is no mention of any quantitative findings at all. It is recommended to the writers that they add some quantitative findings in order to improve the readability and comprehension of the paper.
Response 1: We want to thank reviewer 2 for giving us the opportunity to improve our manuscript and we really appreciate his/her comments. In this sense, we have added some quantitative findings to the abstract that will benefit the comprehension of this work.
Point 2: Why does the author only estimate the impact of CO2 to be 5%?
Response 2: As a preliminary experiment, we tested the impact of different CO2 concentrations on the pH of Chlorella cultures in batch. We found that concentrations above 6% lowered the pH and we decided to keep 5% as the highest concentration in order not to deal with another parameter that could influence our results, especially those related to metabolic changes.
Point 3: How can we determine whether or not a CO2 concentration of 5% meets the criterion for being deemed high?
Response 3: We agree with reviewer 2 that 5%CO2 should not be considered as high CO2, and we have changed that all over the manuscript in order to avoid confusion.
Point 4: The innovation of this study must be emphasized in the introduction.
Response 4: We want to thank reviewer 2 for this suggestion. We have followed this suggestion and we have also added a conclusion section to make more visible the results and the innovation of this study.
Point 5: Line 90-91, This phrase is unacceptable to me due to the fact that the CO2 tolerance ability of microalgae was changed not only by the concentration of CO2 but also by the flow rate of CO2. In addition, a number of studies have shown that Chlorella can thrive in environments with high concentrations of CO2 ranging from 10 to 40%.
Response 5: This phrase has been removed in the new version of this manuscript following reviewer 2 indications.
Point 6: The authors should include a concise commentary on the drawbacks of the systems that were investigated. To provide the reader with a deeper understanding of the topic, it is recommended that one whole paragraph be included.
Response 6: Following this recommendation, we have added a comment in the introduction on the limitations of using inositol polyphosphates in deciphering nodes of regulation in green cells.
Point 7: When writing the abstract, the body of the text, the table, or the figure, you should avoid using abbreviations before explaining what they mean.
Response 7: We appreciate this comment and apologize for this inconvenience. We have added explanations to each abbreviation in the new version of the manuscript.
Point 8: References need update: Although the number of references is already very high, further recent relevant references from the past 5 years should be cited. Please check if you really need all those references cited.
Response 8: Following this indication, we have reduced the number of references and updated them along the manuscript.
Point 9: The manuscript cites 59 references, none of which were published in the PLANTS, 2 suggesting that it is not of direct relevance to PLANTS readers. Consider also citing reference(s) published in PLANTS.
Response 9: Following this indication, we have reduced the number of references and also revised the scope of the journal. In this sense, we still think that this manuscript falls into the scope of PLANTS and specifically in one of the special issues related to carbon metabolism. This submission was also suggested by one of the editors of this journal.

Round 2
Reviewer 1 Report
Thanks to the authors for critically addressing my concerns and I'm satisfied with the current version.
Reviewer 2 Report
Manuscript has been improved agree with Reviewers comments and suggestions. In my opinion manuscript can be publish in current form.